# Safety of Total Knee Arthroplasty without Using a Tourniquet in Elderly Patients

**DOI:** 10.3390/geriatrics6040100

**Published:** 2021-10-16

**Authors:** Satoshi Miyamoto, Masahide Kosugi, Shin Sasaki, Ken Okazaki

**Affiliations:** 1Department of Orthopaedic Surgery, Kohsei Chuo General Hospital, Tokyo 153-8581, Japan; masahidemd@me.com (M.K.); sasaki-shin@kohseichuo.jp (S.S.); 2Department of Orthopaedic Surgery, Tokyo Women’s Medical University, Tokyo 162-0054, Japan; okazaki.ken@twmu.ac.jp

**Keywords:** elderly, total knee arthroplasty, tourniquet, hemodynamics

## Abstract

This study retrospectively compared the perioperative bleeding, hemodynamics, and clinical outcomes of total knee arthroplasty (TKA) performed with and without a tourniquet between two age groups. We grouped 103 patients with knee osteoarthritis who underwent primary TKA based on age at surgery: <76 years and ≥76 years. Tourniquet was used for TKA until March 2010 and stopped thereafter; hence, the patients were further classified according to TKA performed with or without a tourniquet. The differences in the operation time; perioperative bleeding; estimated bleeding; and hemoglobin (Hb) and hematocrit (Ht) levels immediately, 1 day, and 7 days postoperatively were evaluated. The clinical outcomes for range of motion, and Knee Society Knee Scores preoperatively and at 4 weeks postoperatively were assessed. Operation time was longer in the ≥76-year-old non-tourniquet group. No difference was observed in estimated bleeding among the groups. Changes in the Hb and Ht levels at postoperative days 1 and 7 were negatively correlated with age but were not different for TKA performed with or without a tourniquet in the ≥76-year-old-patient group. There were no differences in clinical outcomes among the groups. TKA can be performed with or without a tourniquet in patients aged ≥ 76 years with careful assessment of postoperative anemia.

## 1. Introduction

The number of total knee arthroplasties (TKAs) has been increasing worldwide because of the increased number of people who are older owing to a high life expectancy. Previous studies revealed that people who are older and who undergo TKA require longer hospital stays [1,2,3] and rehabilitation [1] and have a higher risk of postoperative complications [2,3] than younger patients who undergo TKA. Despite these concerns, TKA has significantly contributed toward improving the quality of life of patients [4].

TKA performed without a tourniquet tends to bleed and leads to difficulty in securing the intraoperative field of view [5]. Many studies have examined the risks and benefits of tourniquet use in TKA, and a recent systematic review with meta-analysis revealed that TKA performed with a tourniquet had a higher risk of venous thromboembolic (VTE) events, greater postoperative pain, and longer hospital stays but no difference in overall blood loss compared with TKA performed without a tourniquet [6]. However, few studies have determined the benefit and safety of TKA performed without a tourniquet in patients who are older, such as those aged > 70 years. Patients who are older have a high frequency of comorbidities, including cardiovascular disorders. Although previous studies have shown the safety of TKA without tourniquet use in their study populations, concerns still exist for patients who are older, which has become an increasingly large group in Japan.

In our clinical practice, we stopped using tourniquet for TKA starting from 2010. This retrospective study compared the outcomes in two groups of patients who are older (<76 and ≥76 years) and who underwent TKA with or without a tourniquet. We investigated blood loss, hematological data, range of motion (ROM), and the Knee Society Knee Score (KS score). We hypothesized that there are no differences in blood loss and clinical outcomes among groups undergoing TKA with or without a tourniquet, showing its safety in patients who are older.

Informed consent was obtained from all subjects involved in this study.

## 2. Materials and Methods

### 2.1. Participants

This retrospective study included 135 knees of 121 patients who underwent primary TKA in our department from January 2009 to May 2013. We excluded eight knees of eight patients with rheumatoid arthritis or osteonecrosis, and eight knees of four patients who underwent simultaneous bilateral TKA from this study. Of the remaining 119 knees of 109 patients, 16 knees of 16 patients had insufficient data; thus, 103 knees of 93 patients were investigated (follow-up rate: 86.6%). We used a tourniquet until March 2010 and stopped thereafter. As the mean age of this cohort was 75.8 years, the 103 knees were categorized into two groups: <76 years old (group A: 56 knees, mean age, 68.9 ± 6.1 years (51.9–75.9 years)) and ≥76 years old (group B: 47 knees, mean age, 80.4 ± 3.5 years (76.0–90.5 years)). In addition, patients in this age group were further classified into two groups: those who underwent TKA with a tourniquet and without a tourniquet. Data on age, sex, body weight, body height, the American Society of Anesthesiologists Physical Status Classification (ASA), and use of antiplatelet agents were collected from patient medical records. The demographic data of the patients are shown in Table 1.

### 2.2. Surgical Procedures

All surgeries were performed by one of the two doctors (S.M. and M.K.) who specialize in knee surgery with >15 years of experience in performing a similar procedure. In the tourniquet group, a tourniquet with 280 mmHg pressure was used starting from skin incision to the end of the surgery. In the non-tourniquet group, a tourniquet was not used, and 20 mL of 0.5% epinephrine was injected into the joint immediately before the surgery to prevent postoperative bleeding. Using the medial parapatellar approach with gap-balancing techniques, we prepared both the femur and tibia perpendicular to the mechanical axis. The patella was selectively resurfaced for patients with severe degenerative changes. The implants used were PFC ∑^®^ (DePuy Synthes, Johnson & Johnson, Warsaw, IN, USA), Triathlon^®^ (Stryker Orthopedics, Mahwah, NJ, USA), and Vangard^®^ (Zimmer-Biomet Inc., Warsaw, IN, USA). All of the components were cemented. The bone and implant interface was carefully dried prior to cementing using gauze and suction. After closing the wound, 20 mL of saline solution was injected into the joint, and the drain clamp method was used for 2 h after surgery. All drains were removed on the next day. An autologous transfusion of 400 mL of blood was routinely performed for all patients on the day of surgery unless the patients showed preoperative anemia (hemoglobin (Hb) < 11.0 g/dL or low body weight < 40 kg).

### 2.3. Data Collection

Data on the operation time, intraoperative blood loss, and postoperative blood loss measured in the drain bag were obtained from patient medical records. Perioperative blood loss was defined as the sum of intraoperative and postoperative blood loss. The estimated bleeding volume using Nadler’s equation [7] based on blood sampling data was obtained. The differences in the Hb and hematocrit (Ht) levels between preoperative and postoperative days 1 and 7 were calculated. Adverse events, such as neurological dysfunction or symptomatic deep vein thrombosis (DVT)/VTE, were also recorded. In addition, the ROM and the KS score at the time of admission and 4 weeks after the surgery were assessed.

### 2.4. Statistical Analyses

Mean values with standard deviation (±SD) are described. An analysis of variance was used to compare the four groups, and Tukey’s test was used to assess the differences among the groups. We also used Pearson’s correlation coefficient to assess factors that affect age. All analyses were performed using SPSS^®^ version 25 (IBM, Chicago, IL, USA). A significance level of <5% was used; a correlation coefficient (r) ≥ 0.2 was defined as a positive correlation, and r ≤ 0.2 was defined as a negative correlation.

## 3. Results

### 3.1. Patient Background

Body height was significantly higher in tourniquet group A than in tourniquet group B (*p* = 0.024) and in non-tourniquet group A than in tourniquet group B (*p* < 0.001). ASA was significantly higher in non-tourniquet group B than in tourniquet group A (*p* = 0.013) and in non-tourniquet group A (*p* = 0.050) (Table 1). The use of antiplatelet agents was noted in a few patients in group A, but the usage was not significantly different among the groups.

### 3.2. Perioperative Data

The operation time was significantly longer in non-tourniquet group B than in tourniquet group A (*p* = 0.034) and in tourniquet group B (*p* = 0.011). Perioperative blood loss was significantly greater in non-tourniquet group A than in tourniquet group A (*p* = 0.048). However, there was no clear difference in the estimated blood loss. The autologous transfusion of 400 mL blood was scheduled to be performed for most patients, but some of the patients did not meet the criteria. There was no clear difference in the blood transfusion rate (Table 2).

### 3.3. Changes in Hematological Data

The fluctuation in Hb levels is shown by the change in preoperative- and postoperative-Hb levels. The decrease in Hb at 1 day postoperatively was significantly smaller in tourniquet group A than in non–tourniquet group B (*p* = 0.018, 95% confidence interval (CI): −1.514 to −0.1021); (Figure 1).

The fluctuation in Ht levels is shown by the change in preoperative and postoperative Ht levels. The decrease in Ht at 1 day postoperatively was significant smaller in tourniquet group A than in non–tourniquet group B (*p* = 0.010, 95% CI: –4.674 to –0.4566; Figure 2).

### 3.4. Complications

Symptomatic DVT was treated in one patient in tourniquet group A, in one patient in tourniquet group B, and in two patients in non-tourniquet group B. There were no other serious complications, including major bleeding, delay in wound healing, infection, and VTE. All patients were discharged without any extension of the scheduled hospital stay.

### 3.5. Clinical Evaluations

There was no clear difference in the preoperative and postoperative ROM among the groups. The KS score improved postoperatively compared with preoperatively, with no clear differences among the groups (Table 3).

### 3.6. Correlation with Age

There were negative correlations between age and height, weight, preoperative KS score, and Hb level changes at postoperative days 1 and 7 (Table 4).

## 4. Discussion

In this retrospective study, we compared perioperative and estimated postoperative bleeding and hemodynamics after TKA was performed with or without a tourniquet, particularly focusing on elderly patients aged ≥76 years because of concerns regarding impaired circulatory dynamics in elderly patients. In fact, the ASA classification was significantly high in the group with patients who are older. Nevertheless, no differences were observed in blood loss between the TKA performed with and without a tourniquet in the patients who are older. In addition, the clinical results in ROM and KS were not different among the groups. The present study suggests that TKA without a tourniquet can be safely performed in patients who are older and would provide blood loss and clinical outcomes similar to those in patients undergoing TKA with a tourniquet or those who are younger, although the operation time is longer without a tourniquet.

According to the national registry data, the mean age of patients undergoing primary TKA is 67 years in the US, 70 in the UK, 69 in Sweden, and 68 in Australia [8,9,10,11]. By contrast, the mean age of our cohort was approximately 76 years, which is higher than the mean age of patients in Western countries. The number of patients in their late 70s and 80s has been increasing in our population and among patients undergoing TKA. Therefore, information from studies in Western countries does not always match the information of our patient population. Postoperative results should be predicted according to the effects of aging when considering surgical indications. There are few reports on the upper age limit for TKA. Antonio et al. reported slight differences in postoperative complications, bleeding volume, and outcomes between patients who are older and patients of average age [4], and Murphy et al. reported that the length of stay, postoperative complications, and mortality rates are high risk factors for patients who are older [2]. Michele et al. reported that patient-derived results were nearly the same for all age groups [3]. In the present study, there were no differences in estimated blood loss between the two age groups. Perioperative blood loss was greater without the use of a tourniquet in the younger group and decreases in Hb and Ht levels at 1 day postoperatively were smallest in this group, suggesting that an improved intraoperative hemostasis method is required. Although no clear differences were observed in the groups with patients who are older, decreases in postoperative Hb and Ht levels were relatively high at all time points in patients aged ≥ 76 years. Negative weak correlations with age were observed for changes in Hb and Ht levels at postoperative days 1 and 7. This finding suggests that hemodynamic changes should be monitored closely in patients who are older. Qi et al. reported that Hb and Ht reached their lowest levels at 4 days postoperatively and returned to normal levels within 6–12 days postoperatively [12]. However, neither group recovered at 7 days postoperatively in the present study.

Controversy still exists regarding the difference in risks and benefits between TKA performed with or without a tourniquet. Ozkunt et al. reported that long-term tourniquet use in TKA increases pain and decreases functional recovery [13], whereas Alexandersson et al. reported no clear advantage of non-tourniquet TKA in recovery [14]. Regarding cement fixation, Ejas et al. found that the tibial component could be stabilized using cement fixation even without tourniquet use [15]. Jawhar et al. also showed no differences in periprosthetic cement penetration in their randomized control trial [16]. In contrast, Lu et al. reported a meta-analysis with eight RCTs showing that tourniquet use increases the thickness of bone cement around the implant [17]. Therefore, controversy still exists whether the use of tourniquet affects the fixation stability of the cemented implant, and there are no reliable evidences regarding long-term implant survival [18]. Regarding perioperative blood loss, Jawhar et al. also reported reduced intraoperative bleeding with the use of a tourniquet but no difference in total bleeding volume [16]. They also subsequently reported no difference in functional outcomes and patient-reported outcomes in the same cohort [19]. However, Hasanain et al. reported that the use of a tourniquet reduced total bleeding without postoperative complications [20]. Palanne et al. reported that the use of a tourniquet mildly reduces the decrease in Hb levels [21], and Ledin et al. reported an overt reduction in intraoperative bleeding with the use of a tourniquet but only a slight reduction in hemoglobin dilution 4 days postoperatively relative to total bleeding [22].

With regard to tourniquet use, Wang et al. reported reduced intraoperative bleeding with long-term tourniquet use but reduced postoperative and potential bleeding with short-term tourniquet use [23]. Moreover, short-term tourniquet use was believed to reduce the potential bleeding volume in people who are older. Fan et al. reported no association among surgery time, bleeding volume, and recovery [24]. However, Huang et al. reported that short-term tourniquet use was not favorable for achieving early functional results [25], and Kvederas et al. reported the highest estimated bleeding volume in limited use of a tourniquet only during cement fixation compared with the other longer-use strategies [26]. A systematic review and meta-analysis was recently published that discussed these differences and controversies among many studies regarding the differences in tourniquet use [6]. They concluded that the lack of a tourniquet did not increase total blood loss and perioperative complications. However, there are still concerns about whether similar outcomes would be observed in patients who are older. Therefore, we conducted this retrospective study and found that there were no clear differences in estimated total blood loss and postoperative outcomes among groups that are relatively young and who are older for TKA performed with or without a tourniquet. However, the operation time was significantly longer in the group with patients who are older and without a tourniquet than in the group with patients who are older and with a tourniquet, and in the younger group with a tourniquet, suggesting that intraoperative hemostasis may take even longer in patients who are older undergoing TKA without a tourniquet. We assumed that TKA could be performed without using a tourniquet even in patients who are older if the operation time can be reduced by improving the surgical technique.

This study had some limitations that should be addressed. This was a retrospective study that compared two groups of patients who are older (<76 and ≥76 years old) and who underwent TKA with or without a tourniquet. Therefore, the patients’ background was not matched among the groups. However, the surgeries were performed by two surgeons who utilized the same technique, and postoperative management was not changed. We did not review patient histories, postoperative blood pressure fluctuations, and the presence of asymptomatic DVT. Autologous blood transfusion was routinely performed for the patients unless they met the exclusion criteria. As the rate and volume of transfusion were not different among the groups, they did not affect the present study’s conclusion. However, these results may not be applicable to patients without transfusions. The number of patients in each group was small, and large-scale data could not be extracted. Only knee osteoarthritis was targeted in this study, and other diseases, including rheumatoid arthritis, may need to be considered.

## 5. Conclusions

TKA can be performed safely without a tourniquet in patients aged ≥76 years by closely monitoring postoperative anemia, as shown in the blood loss and postoperative clinical outcomes that were similar to those of the patients who underwent TKA with a tourniquet or to those of the younger patients.

## Figures and Tables

**Figure 1 geriatrics-06-00100-f001:**
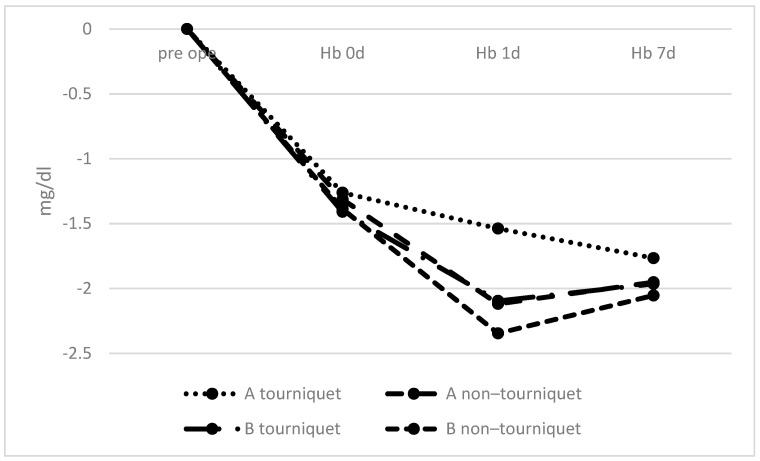
Changes in hemoglobin levels over time. There was a significant difference in hemoglobin levels at 1 day postoperatively between tourniquet group A and non–tourniquet group B.

**Figure 2 geriatrics-06-00100-f002:**
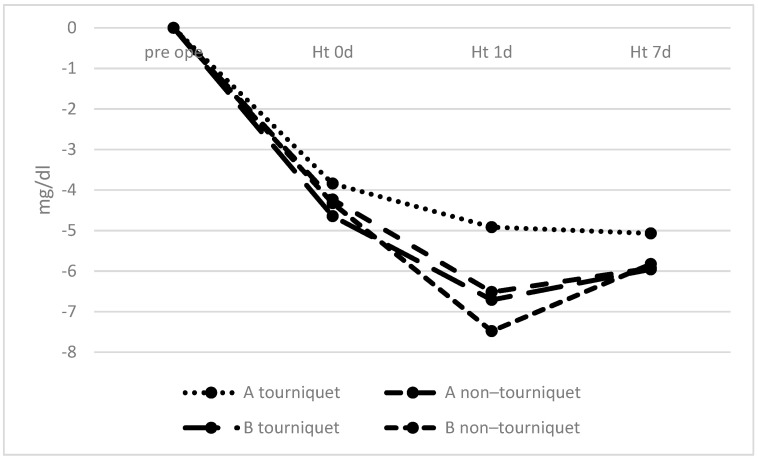
Changes over time in hematocrit levels. There was a significant difference in hematocrit levels at 1 day postoperatively between tourniquet group A and non–tourniquet group B.

**Table 1 geriatrics-06-00100-t001:** Demographic data of the patients. Means ± standard deviations.

Variables	Group A (Age < 76), *n* = 56		Group B (Age ≥ 76), *n* = 47	
Tourniquet group	use, *n* = 29	no use, *n* = 27	use, *n* = 23	no use, *n* = 24
Age, years	68.4 ± 6.6	69.6 ± 5.4	80.1 ± 3.7	80.6 ± 3.2
Sex, male/female	4/25	5/22	1/22	4/20
Body height, cm	152.5 ± 7.7	155.8 ± 8.0	146.4 ± 7.3	151.5 ± 6.9
Body weight, kg	61.4 ± 13.5	64.4 ± 10.8	52.8 ± 8.1	56.0 ± 7.8
ASA, median [IQR]	2 (1–2)	2 (1–2)	2 (2–2)	2 (2–2)
Antiplatelet agents	2/29 (6.9%)	2/27 (7.4%)	0/23 (0%)	0/24 (0%)

ASA: The American Society of Anesthesiologists Physical Status Classification. IQR: interquartile range.

**Table 2 geriatrics-06-00100-t002:** Perioperative data.

Variables	Group A (Age < 76)		Group B (Age ≥ 76)		ANOVA *p* Value
Tourniquet Group	Use	No Use	Use	No Use	
Operation time, min	92.0 ± 15.8 *	99.8 ± 19.2	88.6 ± 15.1 ^†^	108.6 ± 32.5 *^,†^	0.009
Perioperative blood loss, mL	179.2 ± 113.3 ^§^	261.0 ± 127.2 ^§^	183.4 ± 85.3	204.0 ± 96.5	0.046
Estimated blood loss, mL	391.4 ± 151.1	389.7 ± 178.6	393.3 ± 225.2	345.7 ± 165.6	0.770
Transfusions	29/29 (100%)	26/27 (96.4%)	22/23 (95.5%)	22/24 (91.7%)	0.466

*: *p* = 0.034, ^†^: *p* = 0.011, ^§^: *p* = 0.048.

**Table 3 geriatrics-06-00100-t003:** Clinical evaluations.

Variables	Group A (Age < 76)		Group B (Age ≥ 76)		ANOVA *p* Value
Tourniquet Group	Use	No Use	Use	No Use	
ROM (preoperative), extension	10.2 ± 10.0	4.9 ± 8.3	6.7 ± 5.6	6.2 ± 8.8	0.126
ROM (preoperative), flexion	126.8 ± 14.6	116.9 ± 27.5	118.7 ± 22.0	122.1 ± 15.2	0.307
ROM (4 weeks), extension	3.3 ± 5.2	2.0 ± 3.5	1.7 ± 3.6	2.1 ± 4.1	0.529
ROM (4 weeks), flexion	106.9 ± 13.5	109.0 ± 17.4	109.6 ± 9.4	110.6 ± 8.6	0.756
KS score (preoperative), point	52.3 ± 4.7	54.6 ± 10.1	49.3 ± 5.8	50.0 ± 7.6	0.054
KS score (4 weeks), point	74.6 ± 5.8	75.6 ± 6.5	72.7 ± 6.3	75.2 ± 5.4	0.355

**Table 4 geriatrics-06-00100-t004:** Correlation with age.

Age	Pearson’s Correlation Coefficient (r)
vs. Height	−0.269
vs. Weight	−0.400
vs. KS score (pre)	−0.255
vs. change in Hb (1 d)	−0.281
vs. change in Hb (7 d)	−0.22
vs. change in Ht (1 d)	−0.295
vs. change in Ht (7 d)	−0.220

## Data Availability

The data presented in this study are available from the corresponding author upon reasonable request.

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
