# Peer review of "Safety of Total Knee Arthroplasty without Using a Tourniquet in Elderly Patients"

_geriatrics, 2021, doi:10.3390/geriatrics6040100_

Round 1
Reviewer 1 Report
The work has imporved, thanks you for taking my comments into consideration
Reviewer 2 Report
Thank you for addressing my concerns.
This manuscript is a resubmission of an earlier submission. The following is a list of the peer review reports and author responses from that submission.
Round 1
Reviewer 1 Report
Dera sir, the work is interesting but has some drawbacks. The names of the groups should be clarified, A and b for the four groups are inadequate. ASA grade is a categorical variable, not quantitative. An ASA grade of 1.7 cannot exist. Please change it.
Reviewer 2 Report
I am very interested in knowing if the implants were cemented? Lack of tourniquets and poor hemostasis could have major negative consequences on the cement-implant interface. Therefore, the methods section requires clarification on whether or not cement was used and your technique in procedures without tourniquet that were cemented. Then, the discussion section needs to address potential implant stability issues in nontourniquet TKAs.